# Selective Laser Trabeculoplasty in the Treatment of Ocular Hypertension and Open-Angle Glaucoma: Clinical Review

**DOI:** 10.3390/jcm10153307

**Published:** 2021-07-27

**Authors:** Aleksandra Zgryźniak, Joanna Przeździecka-Dołyk, Marek Szaliński, Anna Turno-Kręcicka

**Affiliations:** 1Clinic of Ophthalmology, University Teaching Hospital, ul. Borowska 213, 50-556 Wroclaw, Poland; klo@usk.wroc.pl (A.Z.); marek.szalinski@umed.wroc.pl (M.S.); anna.turno-krecicka@umed.wroc.pl (A.T.-K.); 2Department of Optics and Photonics, Wroclaw University of Science and Technology, wyb. Stanislawa Wyspianskiego 27, 50-370 Wroclaw, Poland; 3Department of Ophthalmology, Wroclaw Medical University, ul. Borowska 213, 50-556 Wroclaw, Poland

**Keywords:** selective laser trabeculoplasty, ocular hypertension, open-angle glaucoma, glaucoma treatment, first-line glaucoma treatment, glaucoma, quality of life, intraocular pressure

## Abstract

Selective laser trabeculoplasty (SLT) is a glaucoma treatment that reduces intraocular pressure (IOP). Its mechanism is based on the biological effects of the selective application of laser energy to pigmented trabecular meshwork (TM) cells, resulting in increased outflow facility. Herein, we review current publications on SLT and summarize its efficacy and safety for different indications in open-angle glaucoma (OAG) and ocular hypertension (OHT) treatment. SLT effectively reduces IOP when used as a primary treatment. In patients whose IOP is medically controlled, SLT helps to reduce medication use, and when maximally tolerated topical therapy is ineffective, SLT facilitates the realization of the target IOP. SLT is a repeatable procedure for which the vast majority of complications are mild and self-limiting. With effective IOP reduction, low complication rates and the potential to repeat the procedure, SLT offers the possibility of delaying the introduction of medical therapy and other more invasive treatment modalities while simultaneously avoiding the accompanying complications. With this knowledge, we suggest that SLT be considered as an essential primary treatment option in OAG and OHT, switching to other treatment modalities only when laser procedures are insufficient for achieving the required target IOP.

## 1. Introduction

Intraocular pressure (IOP) is the only modifiable risk factor in the development and progression of glaucomatous optic neuropathy [1,2]. Therefore, decreasing IOP is a fundamental objective of glaucoma treatment. In addition to pharmacotherapy and incisional surgery, laser therapy has been widely used in glaucoma treatment for many years. In 1995, Latina and Park introduced SLT and proved that laser treatment could be applied selectively to pigmented trabecular meshwork (TM) cells [3]. In contrast to argon laser trabeculoplasty (ALT), the previous standard treatment, SLT avoids thermal damage to adjacent non-pigmented structures [4]. The Food and Drug Agency (FDA) approved the procedure in 2002, and since that time, it has become a widely used treatment option as both a primary and adjunctive treatment for most types of glaucoma. The results of published clinical trials provide a high level of evidence of the efficacy and safety of SLT for several indications. We suggest that SLT be considered as an important treatment option at different stages of glaucoma.

## 2. Methods

The aim of this paper is to summarize key clinical points associated with different indications of SLT used in ocular hypertension (OHT) and open-angle glaucoma (OAG) patients (see Appendix A for the database search strategy and discussion). After removing duplicate studies, we selected 62 published reports for our analysis. We narrowed our review to publications from the past 10 years that address the efficacy and safety of clinical treatment. Additionally, we performed a side search summarizing the possible use of SLT in the closed-angle glaucoma. As this is only a possible introduction for future synthetic research, we just indicated the most important studies and did not perform a full search strategy on this subject.

## 3. Mechanisms of Action

Although SLT has been widely used for years, the exact mechanisms by which it reduces IOP have still not been established. Several studies have shown possible cellular, histopathological and biological effects in TM and Schlemm’s canal cells.

### 3.1. Cellular and Histopathological Changes

In their historical in vitro study in 1995, Latina and Park showed that selective photothermolysis could be applied to TM cells. The researchers focused on the ability of melanin, a chromophore, to absorb laser energy with specific parameters. Mixed cultures of pigmented and non-pigmented TM cells were exposed to a 532 nm Q-switched Nd:YAG laser. The study showed that with a pulse duration of 10 ns to 1 µs, both melanosomes and lysosomal membranes in the pigmented TM cells were disrupted. At the same time, mitochondria lost their structure. These effects were limited to pigmented TM cells. The exposure times used were shorter than the melanin thermal relaxation time, thus avoiding collateral thermal damage to the adjacent non-pigmented TM cells [3,5,6].

A study by Kramer et al. compared post-SLT and post-ALT changes in the uveal meshwork portion of the TM in autopsy eyes. While TM structures were observed to be disrupted with evident coagulative damage (ablation craters) after ALT, SLT-treated tissues showed minimal mechanical damage. Instead, ultrastructural changes, such as the cracking of intracytoplasmic pigment granules and the disruption of TM endothelial cells, were observed [4]. The results of the study by SooHoo et al. were consistent with the work by Kramer et al. in terms of the ultrastructural effects of SLT applied with standard low energies to cadaveric corneal rims. However, tissues treated with energy of 2 mJ showed signs of destruction and scrolling of trabecular beams at the edges of laser burns [7]. Because the lower energies in the SLT method cause almost no coagulative damage, the mechanical effects are minimized. The biological effects that we describe below appear to be a plausible explanation of the SLT mechanism of action.

### 3.2. Biological Changes

SLT has been proven to evoke changes in cellular activity, such as the secretion of different interleukins and metalloproteinases as well as the recruitment of monocytes, leading to the remodeling of the juxtacanalicular extracellular matrix and increasing outflow facility.

Some studies on ALT mechanisms have shown laser-induced biological effects in TM cells. Parshley et al. hypothesized that a turnover of the trabecular extracellular matrix that is regulated by metalloproteinases plays a crucial role in regulating aqueous humor outflow. Among the metalloproteinases secreted by TM cells, stromelysin-1 (metalloproteinase-3, MMP-3) has the broadest substrate specificity. It degrades the globular domains of proteoglycan core proteins, laminin, fibronectin, type IV collagen and other proteins. Researchers evaluated the MMP-3 activity and levels as well as its mRNA levels in laser-treated organ cultures that mimicked the human TM. The study showed a several-fold increase in all parameters at different time points after laser trabeculoplasty. These increases were mainly observed in the insert and juxtacanalicular regions of the meshwork. Bradley et al. showed that MMP-3 secretion in the TM after laser treatment is mediated by increased secretion of the inflammatory interleukins IL-1β and TNF-α [8,9]. SLT may induce similar effects in the TM.

Lee et al. observed an increase in MMP-3 secretion from co-cultures of pigmented and non-pigmented human TM cells after SLT. Interestingly, isolated non-pigmented TM cells showed no response to SLT treatment in terms of MMP-3 secretion. Conversely, MMP-3 secretion from isolated pigmented TM cells declined after SLT. A possible explanation is that laser-treated cells that undergo cell death stimulate adjacent non-treated cells to secrete MMPs. This study also examined the SLT effects on cellular activity and revealed a decline in cellular metabolic activity and an increase in necrosis and apoptosis after SLT in the tested co-cultures [10].

Another biological effect of SLT is the recruitment of monocytes, which influence outflow facility. Alvarado et al. observed an increased number of monocytes in human and monkey eyes treated with SLT. To investigate the effects of their recruitment, autologous macrophages were infused into the anterior chamber in rabbits, and an increase in outflow facility was observed. Additionally, this study showed that human monocytes and monocyte-secreted factors induced increased conductivity in human Schlemm’s canal endothelial cells in vitro [11]. In another in vitro study, Alvarado et al. showed that the laser-induced increase in the permeability of Schlemm’s canal cells was due to intercellular junction disassembly [12].

The main mechanisms described in the literature are summarized in Figure 1.

## 4. Technique

SLT is performed under topical anesthesia with the use of goniolens. A frequency-doubled q-switched 532 nm Nd:YAG laser with a pulse duration of 3 ns and spot size of 400 µm is utilized. A laser beam is focused on the entire width of the TM, and the laser energy is titrated from 0.3 mJ until bubble formation becomes visible in the treatment area. Once the bubble is visible, the laser energy is decreased by 0.1 mJ until the bubble effect is minimized. Some clinicians prefer to see bubbles in only some of the spots. In the trials that we reviewed, the energies varied from 0.2 to 1.7 mJ. Once the appropriate energy is established, treatment is continued over 90, 180 or 360 degrees of the TM. In principle, 25 spots should be applied for every 90 degrees of the TM [5,13,14,15].

Since the introduction of the technique, several protocols concerning the treatment degree of the TM and the number of spots have been evaluated. Table 1 summarizes differences in the effectiveness of SLT in relation to the scope of the procedure performed (the number of quadrants of the circuit covered by laser spots).

## 5. Postoperative Treatment

Depending on the clinician’s practice, short-term anti-inflammatory and IOP-reducing topical treatments may be used after SLT. We found several studies evaluating the impact of postoperative treatment on the efficacy and safety of SLT.

### 5.1. Anti-inflammatory Therapy

The “Steroid After Laser Trabeculoplasty (SALT)” trial was a double-masked, randomized, placebo-controlled trial that evaluated the impact of post-SLT medication on IOP reduction in POAG, exfoliative glaucoma (XFG) and OHT patients. The study was performed on 96 eyes of 85 patients, who were randomized into three groups for treatment with 0.5% ketorolac, 1.0% prednisolone or saline tears four times a day for 5 days after SLT. Twelve weeks after the SLT procedure, IOP had decreased by −6.2 ± 3.1 mmHg, −5.2 ± 2.7 mmHg and −3 ± 4.3 mmHg in the ketorolac, prednisolone and placebo groups, respectively. The differences between the treatment and placebo groups were statistically significant. Post-SLT treatment with prednisolone or ketorolac proved to positively contribute to the IOP-lowering effect [23].

The results of the SALT study contradict those of a similarly constructed study by Jinapriya et al., who found no statistically significant difference between groups treated with 1.0% prednisolone acetate, 0.5% ketorolac tromethamine and artificial tears at the one-month follow-up. In this study, SLT was less effective, and the most probable reason was a low baseline IOP [23,24]. Similar results were shown in a prospective randomized trial by de Keyser et al., who performed SLT in both eyes of 66 patients. One eye received 0.1% indomethacin or 0.1% dexamethasone, whereas the other did not receive any anti-inflammatory treatment. The researchers found no statistically significant difference in terms of IOP reduction or topical inflammatory reactions between groups at the 6-month follow-up [25,26]. Similarly, in a retrospective review, Rebenitsch et al. compared the use of loteprednol for 5–7 days after 360° SLT against treatment without loteprednol and observed no statistically significant difference in the absolute or percentage decrease in IOP [27].

### 5.2. Preventing IOP Spikes

IOP spikes are common adverse events that occur shortly after the laser trabeculoplasty procedure. Several studies have been conducted on the prevention of elevated postoperative IOP. An in-depth review by Zhang et al. was published in 2017, which included 22 randomized clinical trials, with 2112 participants in total. The review indicated that perioperative medication prevented IOP spikes within 2–24 h after laser trabeculoplasty. The study also demonstrated that alpha-2-agonists and pilocarpine were effective. The certainty of the evidence was mainly graded moderate to low [28,29].

### 5.3. Conclusions

Taking the available literature into consideration, it is not possible to establish clear recommendations for post-SLT medication. In terms of long-term IOP-lowering effects, patients might benefit from short-term therapy with prednisolone acetate or ketorolac. Apraclonidine, brimonidine and pilocarpine are significantly more effective than a placebo in preventing postoperative IOP spikes.

## 6. Efficacy of Treatment

The efficacy of SLT in OAG and OHT treatment has been evaluated in a number of trials, which can be divided in accordance with the indications for which the SLT was used.

### 6.1. SLT as First-line Therapy

Published in 2019, the Laser in Glaucoma and ocular HyperTension (LIGHT) study was a multi-center, unmasked, randomized trial that evaluated and compared the health-related quality of life (HRQoL), clinical efficacy, cost-effectiveness and safety between SLT and IOP-lowering eye drops. The subjects were newly diagnosed treatment-naïve patients with OAG or OHT who were monitored over a 3-year period. There were 356 patients randomized in the laser-first group and 362 patients assigned to the medication-first group. For patients with OHT, success was defined as IOP < 25 mmHg and >20% IOP reduction. For OAG patients, unlike in other studies, which have often defined the success rate solely as >20% reduction in IOP from the baseline, a target IOP was established at the beginning of the study for each patient in accordance with glaucoma severity. During the observation, target IOP was reduced, in cases that manifested deterioration despite IOP being at or below target. In cases that remained stable despite IOP being slightly above target, target IOP was revised and higher values at which no deterioration was observed were accepted. Follow-up intervals and excess treatment were determined on the same basis. During the observation period, decision-making about subsequent treatment steps was supported by a defined protocol and digital tool. The study was meticulously designed to make the results both more accessible to clinicians and easy to use in real-life situations. In the laser-first group, patients underwent SLT, which could be repeated once during follow-up. If IOP control was insufficient, patients were further treated with medication. In the medication-first group, patients were treated with topical medications in a stratified manner, whereby a prostaglandin analogue, beta-blocker, topical carbonic anhydrase inhibitor and alpha-adrenoceptor agonist were offered as the first, second, third and fourth treatment choice, respectively [13,14].

HRQoL was assessed using the EuroQol Five Dimension Five Level Scale (EQ-5D-5L) questionnaire, which evaluates five dimensions of quality of life: mobility, self-care, usual activities, pain and discomfort, and anxiety and depression. At 36 months, the score was not significantly different between the groups (adjusted mean difference (laser first–medicine first) 0.01, 95% CI −0.01 to 0.03; *p* = 0.23) [30]. Of the eyes treated with primary SLT, 95% reached the target IOP by the end of the follow-up period. Of this group, 78.2% of the eyes required no additional medication, which showed that 74.2% (95% CI 69.3–78.6) of eyes in the laser-first arm achieved eyedrop-free IOP control for at least 3 years. In the medication-first group, 93.1% of eyes were at the target IOP at 36 months, and of these eyes, 64.6% required only one type of medication. In the laser-first group, the number of treatment escalations, as well as the number of required IOP-lowering and cataract surgeries, was lower than in the medicine-first arm. The difference was critical in the number of trabeculectomies (0 in laser-first arm vs. 11 in medication-first arm). Post-SLT patients did not report any sight-threatening adverse events. After the procedure, only six IOP spikes (>5 mmHg) were observed, whereas transient discomfort, blurred vision, photophobia and hyperemia were reported by 34.4% of patients. Systemic adverse events were comparable between the laser-first and medication-first groups. SLT resulted in a greater quality-adjusted life-year gain at a lower cost than medical therapy, but the difference was not significant (*p* = 0.286) [13,14].

The trial proved SLT to be a clinically effective, cost-effective and safe alternative to eye drops as primary therapy in patients with OAG or OHT. In conclusion, the researchers stated that SLT should be offered as first-line therapy, supporting a change in clinical practice. Due to its exceptional design, which enabled the replication of real-world practice while tailoring the therapy to the patient, the LIGHT study is of outstanding importance in clinical practice [13,14]. Notably, the use of SLT as first-line therapy substantially lowers the non-compliance risk rate and decreases the rate of side effects, thus improving the probability that patients are able to achieve the prespecified target IOP.

In a post hoc analysis of the LIGHT study, Garg et al. compared the clinical efficacy of primary SLT in treatment-naïve patients with OAG and OHT. The researchers reported that primary SLT was a comparably effective IOP-lowering treatment in OAG and OHT patients [25,31].

The results of the LIGHT trial were included in a meta-analysis by Chi et al. with two other trials in treatment-naïve patients, and similar results were observed in terms of the IOP reduction efficacy of SLT [32,33,34].

The results of the LIGHT study contradict those of the Glaucoma Intensive Treatment Study (GITS) trial by Ang et al., which was published in 2020. It compared SLT with escalated medication in POAG and XFG patients, and a better response in terms of decreased IOP was observed in the Medication group at the 12-month (62.3% vs. 45.5%) and 24-month follow-up (72.1% vs. 53.4%). At the end of the observation period, the success rate of medication was 18.6% (95% CI 3.0%–34.3%, *p* = 0.022), which was higher (as an absolute difference) than that in the SLT group. Compared with the LIGHT study, the sample size of GITS was smaller. With 167 patients randomized, GITS did not reach the target size of 386 patients. Treatment success was defined as >25% IOP reduction from the baseline. In contrast to the LIGHT study, in which SLT covered 360° of the TM, in the GITS trial, SLT was applied to 180° of the TM [13,14,15].

An interesting outcome was achieved in Kiddee and Atthavuttisilp’s assessment of the post-SLT reduction in diurnal IOP fluctuations compared with the effect of medication (travoprost) in POAG and normal-tension glaucoma (NTG). In this randomized single-masked study, SLT was introduced as first-line therapy, and the authors did not find any statistical difference in IOP reduction (median reduction in IOP was 3.7 mmHg and 4.1 mmHg in SLT and Travoprost groups, respectively) between groups according to the type of glaucoma. The main outcome, which was success in reducing fluctuations, was achieved in 75% and 92% of subjects in the SLT and Travoprost groups, respectively. In addition, the rate of reduction in diurnal fluctuations was lower in the NTG group than in the POAG group [35].

In a recently published retrospective study by Ansari et al., SLT was found to effectively reduce IOP in POAG treatment-naïve patients over a 10-year observation period. The trial evaluated 108 eyes of 54 patients. With SLT repeated as required, the success rate of treatment, defined as both >20% IOP reduction compared with the pre-treatment value and IOP < 19 mmHg, was 98% at year 1, 89% at year 5 and 72% at year 10. Treatment failure was most common at 3 years. During the observation period, 60% of patients required additional SLT procedures [36].

In summary, we consider SLT to be a promising first-line therapy option that can postpone the need to administer topical medications that may, on the one hand, cause allergic reactions and, on the other, cause changes in the anterior segment of the eye or the entire eye socket [29,37,38,39]. Patients are inclined to consent to treatments that provide an option to be medically independent. In this respect, SLT appears to be an ideal solution since the balance between benefits and complications is acceptable and favorable.

### 6.2. Replacement Therapy

The vast majority of trials have focused on post-SLT reduction in IOP. Fewer studies have evaluated the potential of SLT to reduce topical medication as the main outcome measure. One of the main concerns of glaucomatous patients, however, is the number of medications that they are expected to self-administer on a daily basis. Additionally, non-compliance is largely due to the inadequate administration of eye drops, an inability to adhere to a strict regimen and multiple formulations or an unwillingness to be treated with eye drops. Patients tend to prefer single or even multiple laser procedures if they will “free” them from eye drops.

A study by Francis et al. evaluated 66 patients with controlled POAG and XFG on topical medication. The primary outcome measure was the reduction in the number of medications after SLT over the course of a one-year observation period. The mean reduction in medications from the baseline was 2.0 (1.8–2.3) at 6 months, which represents 95% confidence, and 1.5 (1.27–1.73) at 12 months (*p* < 0.0001) [19,40].

A prospective, comparative, interventional case series by Tufan et al. was designed to assess the medication-reducing effect of SLT in one eye compared with the fellow eye, which continued treatment with a fixed combination with timolol. In the study groups, 22 eyes underwent 180° SLT, and 18 eyes received 360° SLT. Forty fellow eyes made up the control group. At the 6-month follow-up, the study showed no statistically significant differences between groups (*p* < 0.001) [41].

In 2018, de Keyser et al. published the results of their prospective randomized clinical trial on 286 eyes of 143 OAG and OHT patients on topical medication with controlled IOP who received SLT as replacement therapy. Patients were randomized into two groups: an SLT group and a control group that continued topical medication. All patients who underwent SLT reduced their medication, of which 77% needed no medication after 12 months of observation, and 74% no longer needed medication after 18 months. The mean reduction in medication was 1.15 after 12 months and 1.21 after 18 months [42].

In another trial, de Keyser et al. assessed changes in the quality of life of patients after the use of SLT as replacement therapy. The SLT group consisted of 64 patients, and the control group included 61 patients that continued medical therapy. Groups were followed for at least 6 months. While the mean IOPs remained unchanged in both groups during follow-up, the mean number of medications in the SLT group was reduced from 1.56 to 0.42 and 0.33 at 6 and 12 months, respectively. In the SLT group, punctate keratitis was observed in 35.94% of patients at the baseline, 14.06% at the 6-month follow-up (*p* = 0.14) and 12.24% at 12 months. The quality-of-life questionnaire included 15 questions about the perceived effectiveness of the treatment, its side effects, eye appearance changes, its convenience and the ease of administration of the eye drops. One year after SLT, all parameters were rated better by the treatment group than by the control group, with the difference being significant (*p* < 0.001). In this study, SLT was proven to be effective in reducing the number of medications, which resulted in an improvement of the treatment-related quality of life [43].

In summary, SLT is able to “free” patients from topical medication in approximately 70% of cases for 12–18 months after therapy, which results in an improvement of the quality of life and a reduction in topical complications.

### 6.3. Adjunctive Therapy

Besides reducing the number of drugs used in patients with well-controlled IOP, SLT may help to achieve IOP control in patients in whom medical therapy is insufficient. Patel et al. conducted a retrospective review of 67 glaucoma and ocular hypertension patients who had uncontrolled IOP on maximally tolerated medical therapy and underwent SLT as adjunctive therapy. Eight eyes that did not achieve the target IOP in the first 3 months of observation were excluded from the analysis. With success defined as either IOP reduction > 20% from the baseline or reduction in medication without additional laser or surgical intervention, the success rates were 62%, 50% and 32% after 1, 3 and 5 years, respectively. The mean IOP (baseline 18.7 mmHg) was significantly reduced in 4–8 weeks (15.6 mmHg, *p* < 0.001) and at the 1-year follow-up (16.8 mmHg, *p* = 0.005). Similarly, the mean number of medications was reduced significantly at 1-year, 3-year and 5-year follow-up (*p* < 0.001, *p* < 0.001 and *p* = 0.039, respectively) [44].

A UK study that monitored the real-world outcomes of 831 SLT-treated eyes included an assessment of the efficacy of SLT in patients that used prostaglandin analogues. The study showed no difference in SLT failure between 449 patients using prostaglandin analogues and 75 patients using other medications (HR, 0.95; 95% CI, 0.70–1.30; *p* = 0.76) or 382 patients who did not use prostaglandin analogues (HR, 0.95; 95% CI, 0.8–1.12; *p* = 0.56). Additionally, the researchers compared the IOP-lowering effect between 237 patients using prostaglandin analogues and 202 patients not using prostaglandin analogues whose data were available at the 12–18 month time point. The absolute reduction in the prostaglandin analogue group and the non-prostaglandin group was 3.6 mmHg (95% CI, 2.9–4.2; *p* < 0.0001) and 4.9 mmHg (95% CI, 4.2–5.6; *p* < 0.0001), respectively. After adjusting for pre-SLT IOP, the difference was not significant (*p* = 0.81) [45].

The available literature shows that SLT has the potential to reduce IOP in patients whose IOP is uncontrollable with medication alone. The type of medication used before the procedure does not influence the rate of IOP reduction.

### 6.4. Young Patients

Because the IOP needs to be lowered throughout an individual’s lifetime and the risk of intolerance to topical medication evolves with time, SLT can be an attractive option for patients with juvenile open-angle glaucoma. We found only two studies that evaluated SLT in patients under 40 years of age.

Gupta et al. observed 30 eyes of 30 patients diagnosed with juvenile open-angle glaucoma before the age of 40. Patients on maximally tolerable medical therapy and IOPs above the target underwent SLT and were followed up for 12 months. The researchers found that IOP was reduced from 25.3 ± 6.5 at the baseline to 17.3 ± 5.8 mmHg at 12 months (*p* = 0.01). Success was defined as >20%, and IOP reduction was achieved in 43% of eyes at the end of the observation period. The results showed that 23% of eyes presented an IOP reduction of 44%. In 20% of eyes, one medication was reduced. The researchers found no difference in age or baseline IOPs between patients who successfully responded to the procedure and those who did not. The only significant parameter was the presence of goniodysgenesis. Eyes without goniodysgenesis had a 4.3-fold (95% CI 1.1–15.2) higher chance of appropriate IOP reduction compared with those affected by angle dysgenesis (*p* = 0.034) [46].

Liu et al. conducted a retrospective study of 56 eyes of patients under 40 years of age with POAG and OHT in comparison with 23 eyes of patients over 60 years old. All eyes underwent SLT and were followed up for at least 12 months. While younger patients had significantly higher baseline IOPs (*p* = 0.02), IOPs at the 12-month follow-up were not significantly different between groups (*p* = 0.59). Success was defined as IOP reduction > 20% with no change in medical treatment and no need for surgery. The researchers showed success rates of 71.4% and 56.5% in young and elderly patients, respectively. The difference was not significant [47].

In both studies, SLT proved to be effective in young patients with open-angle glaucoma. Since shortening the period of topical medication use or achieving independence from it is of particular value to younger patients, further randomized controlled trials need to be promptly conducted to provide a higher grade of evidence [46,47].

### 6.5. Exfoliative Glaucoma (XFG)

XFG is the most common secondary glaucoma that has a proven worse prognosis and faster progression. Due to higher angle pigmentation that, theoretically, could result in higher efficacy, SLT may appear to be an attractive option in this group of glaucoma patients.

A small prospective, non-randomized study by Shazly showed that the efficacy of SLT as primary therapy was similar in POAG and XFG. At the 30-month follow-up, the mean IOPs in POAG and XGF groups were 17.6 ± 2.8 mmHg and 18.3 ± 4.7 mmHg, respectively. The mean reduction in these groups was 5.7 ± 2.1 mmHg and 5.3 ± 3.0 mmHg, respectively. The slightly lower IOP in the POAG group at the end of the observation period was not statistically significant [48].

Part of a retrospective study conducted in Sweden, known for higher rates of XFG among OAG patients, compared the efficacy of SLT between XFG and POAG patients for different indications (primary treatment, intolerance of medication, and attempt to delay other invasive treatments). The XFG and POAG groups consisted of 114 and 142 patients. The mean baseline IOP was 24.2 ± 5.4 mmHg and 23.3 ± 5.9 mmHg in POAG and XFG groups, respectively. IOP at follow-up was 19.9 ± 5.4 mmHg in the XFG group and 19.8 ± 6.7 mmHg in the POAG group. In both groups, the reduction was statistically significant (*p* < 0.001). The difference in IOP between POAG and exfoliative glaucoma patients at the baseline (*p* = 0.20) and follow-up (*p* = 0.26) was not statistically significant [49].

Similarly, Miraftabi et al. compared 20 XFG with 28 POAG eyes in a prospective study and obtained similar results in terms of the efficacy of IOP reduction in both groups at the 12-month follow-up. In their study, the percentage of IOP reduction was higher in the XFG group than in the POAG group at 6 months, and this difference was statistically significant (*p* = 0.02). This might have resulted from the higher IOP baseline in the XFG group [50].

A retrospective review of 48 POAG and 37 XFG eyes reported success rates of 54.2% and 78.4% in POAG and XFG groups at 12 months, respectively (*p* = 0.039), with SLT success rates defined as IOP ≤ 21 mmHg, reduced medical usage and a ≤ 20% reduction in IOP without additional medications. With a baseline IOP that did not differ significantly, IOP at 12 months was higher in the POAG group, and the difference was statistically significant (*p* < 0.0001). After one year, the mean IOP reduction in POAG and XFG groups was 4.4 ± 2.1 mm Hg and 6.1 ± 3.6 mmHg, respectively. [51].

SLT appears to be an appropriate treatment modality in exfoliative glaucoma, proving as effective in this condition as it is in primary open-angle glaucoma.

### 6.6. Pigmentary Glaucoma

Patients with pigmentary glaucoma (PG) were included in OAG groups in a number of studies, but most of them appeared to be isolated cases. We found only one retrospective study that assessed the efficacy of SLT exclusively in PG patients. The study group consisted of 30 eyes of 30 PG patients. The primary outcome measure was time to failure after SLT, defined as any of the following: <20% IOP reduction, change in medication, and repeated SLT or the need for glaucoma surgery. Time to failure was 27.4 (SD 13.61) months on average. Success rates measured by the Kaplan–Meier curve were 85%, 67%, 44% and 14% after 12, 24, 36 and 48 months, respectively. Researchers reported two cases of postoperative IOP spikes (>6 mmHg), which had diminished by the day after treatment without any change in medication. Further studies might help to clarify differences (if any) in efficacy and safety between PG and other types of glaucoma [52,53,54,55].

### 6.7. Advanced Glaucoma

Although trabeculectomy remains the “gold standard” in treating advanced glaucoma, decision-making about such invasive therapy in most clinical situations is not straightforward. In the majority of cases, clinicians need to consider the relatively high perioperative risk, especially when a reoperation is involved. As a safe and noninvasive treatment, SLT can thus be considered an option. We found two studies that showed promising results.

In their retrospective chart review of 44 eyes (44 patients), Schlote et al. assessed the efficacy of SLT in patients with subtotal papilla excavation and ≥stage 3 in the Glaucoma Staging System 2 scale who underwent SLT because they had insufficient IOP control after treatment, suffered from allergies, experienced discomfort with topical medication or were non-compliant with topical treatment. For this group of patients, four success criteria were defined: Reduction in IOP to <21 mmHg and >20% compared with the baseline (achieved in 26 eyes, 59.1%);IOP reduction to <18 mmHg with no additional medication at all time points after SLT (achieved in 29 eyes, 65.9%);IOP reduction to <18 mmHg and >30% of baseline (achieved in 22 eyes, 50%);The number of eyes that underwent incisional surgery within 12 months after SLT (occurred in 8 eyes).

In this group, the mean baseline IOP was 22.1 ± 4.1 [16,35] mmHg, while the IOP at the 12-month follow-up was 14.8 ± 2.4 [10,26], with the difference being statistically significant (*p* < 0.0001). Eyes that underwent trabeculectomy were excluded from this statistic. The number of medications remained unchanged in the 12-month follow-up period [56].

Sharpe et al. retrospectively compared the efficacy of SLT in 53 eyes with prior trabeculectomy with an ExPress mini shunt, Ahmed valve or combined phacoemulsification-trabeculectomy (prior glaucoma surgery, PGS group) with a matching group of 53 eyes with no prior glaucoma surgery (NPGS group). The indication for SLT was uncontrolled IOP despite maximally tolerable medical therapy. The mean pre-SLT IOP was 19.2 ± 4.3 mmHg in the PGS group and 20.6 ± 6.0 mmHg in the NPGS group. The mean IOP reduction was statistically significant in both groups at the 1- and 6-month follow-up (*p* < 0.04) and reached 7.3% in the PGS group and 10.8% in the NPGS group (*p* = 0.42) at 6 months. The success rate, defined as >20% IOP reduction, was achieved in 28.3% and 24.5% at 1 month and 27.9% and 31.7% at 1 year in PGS and NPGS groups, respectively. In the PGS group, IOP reduction was significantly higher in eyes with a baseline IOP ≥ 21 mmHg compared with those with IOP < 21 mmHg at all follow-ups. In this study, success rates were significantly lower than in other studies evaluating SLT efficacy due to the advanced glaucoma stage, in which TM is heavily dysfunctional. This study shows that, among advanced glaucoma eyes, there is a group in which TM demonstrates some residual activity that can be moderated with laser therapy [57]. 

In contrast to this study, Zhang et al. reported a similarly defined SLT success rate of 77.7% at 9 months in patients with prior trabeculectomy and uncontrolled IOP. With 18 eyes of 16 patients, the study sample was significantly smaller than in the previously described study, and there was no control group [58].

### 6.8. Angle Closure

Although angle closure and angle-closure glaucoma are beyond the scope of this review, it is worth mentioning few studies with promising results that we also found in this indication. 

A randomized clinical trial from 2015 compared efficacy of SLT (49 patients) versus travoprost 0.004% (47 patients) in reducing IOP in patients with primary angle closure (PAC) or primary angle-closure glaucoma (PACG) who underwent laser peripheral iridotomy (LPI) and in whom the angles opened in more than 180°. At 6 months follow-up study showed similar absolute reduction of IOP in the SLT and the travoprost group (4.0 vs. 4.2 mm Hg, respectively; *p* = 0.78). Success defined as IOP < 21 mm Hg and no additional medication was achieved in 60% of the eyes in the SLT group and 84% in the travoporst group (*p* = 0.008). The study proved the SLT to be effective in reducing IOP in PAC and PACG patients in short term observation [59].

A case control study from 2016 compared the efficacy of the SLT in patients with PAC or PACG in whom the angles opened in at least 180° after LPI versus POAG patients. Groups were matched for age, baseline IOP and severity of glaucoma. Patients were followed up for 10 months in the PAC/PACG group and 11 months in the POAG group. At the end of follow-up, the difference in the postoperative IOP reduction was not statistically significant (*p* = 0.66). Success defined as IOP reduction of ≥20% from baseline or reduction in medication of 1 or more drugs was achieved in 84.7% patients in PAC/PACG group and 79.6% in POAG group (*p* = 0.47). The study showed similar efficacy of the SLT in reducing IOP in PAC/PACG patients and POAG patients [60].

A prospective study from 2019 assessed efficacy of the SLT in PAC/PACG patients with the angles opened in at least 180° after LPI versus POAG patients, not only in terms of IOP reduction but also in terms of preventing glaucoma progression in a longer follow-up (6 years). The study showed similar, statistically significant IOP reduction in both groups during entire follow-up period. Glaucoma progression was assessed with the use of guided progression analysis (GPA) on Humphrey Field Analyzer II together with peripapillary retinal nerve fiber layer (RNFL) and ganglion cell complex (GCC) on FD-OCT. The study showed a significant decrease in rates of progression (ROP) according to GPA, RNFL and GCC in both groups after SLT. According to GPA, ROP was significantly faster in PAC/PACG group than in POAG groups in period between 2 to 6 years after SLT. However, the study revealed no such difference according to RNFL and GCC trend analysis. In this study, SLT was proved to be effective in preventing glaucoma progression both in PAC/PACG and POAG patients [61].

## 7. Predictive Factors

As mentioned above, SLT appears to be a generally effective procedure. However, in each trial, there was a group of patients who did not respond to the treatment. For this reason, several trials have been conducted in search of factors that enable the prediction of success. With 72 patients evaluated at their one-year follow-up, Hodge et al. demonstrated that SLT success was strongly predicted by baseline IOP (odds ratio = 1.16; *p* = 0.0001) and not by age, sex, glaucoma type or grade of TM pigmentation [62]. Similar results were shown in a recent retrospective review by Hirabayashi et al. [63]. In this study, at 6 months, patients with a baseline IOP > 18 mmHg showed an IOP reduction of 5.4 ± 5.3 mmHg (23.7% reduction), whereas patients with a baseline IOP < 18 mmHg showed a reduction of −0.7 ± 4.6 mmHg (4.9% increase; *p* < 0.001). Similarly, in this study, SLT success was not significantly predicted by age, type and severity of glaucoma, PTM or total energy delivered. These results are consistent with three other trials. The results of a post hoc analysis of the LIGHT study by Garg et al. (mentioned above in the subsection “SLT as first-line therapy”) showed that early absolute IOP reduction was positively correlated with baseline IOP and negatively correlated with female gender [13,14,31,62,63,64,65,66].

By contrast, a Chinese trial showed that, in addition to high pre-SLT IOP (coefficient = 0.3; OR: 1.3; *p* = 0.0005), older age (coefficient = 0.1; OR: 1.1; *p* = 0.0003), the use of four types of antiglaucoma medication (coefficient = 2.1; OR: 8.4; *p* = 0.005), a higher spherical equivalent (coefficient = 2.1; OR: 8.4; *p* = 0.005) and the use of a topical carbonic anhydrase inhibitor (coefficient = 1.7; OR: 6.0; *p* = 0.003) were significantly associated with success. Additionally, in another study conducted by the same researchers in normal-tension glaucoma patients, the use of three types of IOP-lowering eye drops prior to SLT was negatively associated with success rate (*p* = 0.02) [67].

In summary, the independent predictor appears to be an elevated IOP, itself achieving a reduction of 23–30%. This result is comparable to the effect of topical medication and, as such, can be considered among treatments with the highest rates of IOP reduction.

## 8. Repeatability

Because the energy applied in SLT is relatively low with limited tissue damage, it is a repeatable procedure. Several trials have evaluated the efficacy and safety of repeated SLT.

Hong et al. published a retrospective chart review that analyzed the efficacy of repeated SLT (SLT 2) compared with primary SLT (SLT 1) in 44 eyes of 35 patients with POAG, XFG and PG at several time points. The study group consisted of patients older than 18 years of age whose glaucoma was not adequately controlled with maximally tolerable medical therapy and whose initial SLT procedure was successful (>20% IOP reduction) for at least 6 months. Patients underwent a repeated SLT once the effect of the initial SLT wore off. The only statistically significant difference in IOP after SLT 1 and SLT 2 was the mean change at 1–3 months, which was −5.0 mmHg for SLT 1 and −2.9 mmHg for SLT 2 (*p* = 0.01). Despite its retrospective character and small study group, which included both eyes of several patients to achieve the required sample size, the study demonstrated that SLT can be repeated with success after the effect of the primary SLT wears off [68]. 

Another retrospective chart review by Polat et al. was similarly designed, but only one eye of 38 patients was included. The initial SLT resulted in significant IOP reduction at all follow-up observations in the 24-month period (the mean baseline IOP was 21.6 (4.8) mmHg vs. mean IOP at months 1–24, ranging from 15.9 to 18.6 mmHg (*p* < 0.05)). The mean baseline IOP of repeated SLT was 19.1 (3.9) mmHg. This value was lower than that of the primary SLT because the decision about repeated SLT was made before the effect of the primary SLT was fully depleted. The mean IOP after the repeated SLT in the 24-month observation period ranged from 14.7 to 17.0 mmHg and was significantly reduced from the baseline. To evaluate the median survival time, a Kaplan–Meier survival analysis was conducted, which showed a median survival time of 270 days (9 months) and 360 days (12 months) for the initial and repeated SLT, respectively, when success was defined as >20% IOP reduction. This review showed that both initial and repeated SLT are effective in reducing IOP and that the IOP reduction achieved after the initial SLT can be restored with repeated SLT [69]. 

A larger sample size of 137 eyes of 137 patients and stricter success criteria were advantages of a similarly designed multi-center study by Francis et al. In the first definition, success was defined as an IOP value ranging from 5 to 21 mmHg and IOP reduction from baseline of >20% with no additional glaucoma medications or procedures. The second definition of success was less stringent: an IOP value of 521 mmHg, no glaucoma procedure needed and either an IOP reduction of >20% from baseline or a reduction in medication. After SLT 1, 55% and 35% of eyes met the success criteria included in the first definition at 6 months and 12 months, respectively. With the second definition, the success rates were 65% and 44% at 6 and 12 months, respectively. Post-SLT 2 success rates with the first definition were 37% at 6 months and 19% at 12 months. When rated using the second definition, these values were 48% and 27%, respectively. The larger sample size enabled a sub-analysis in which the success rates of repeated SLT were evaluated based on the time between the initial and repeated SLT. With either definition, repeated SLT performed less than one year after the initial SLT proved to have better success rates than repeated SLT performed more than one year after the first SLT [19].

The repeatability of primary SLT was evaluated in a retrospective chart review by Avery et al. and in a post hoc analysis of the SLT treatment arm in the LIGHT study by Garg et al. [31,70]. 

The first study included 42 eyes of 42 patients who had primary SLT that was repeated when failure of the first treatment was found. Success was defined as an IOP reduction of ≥20% of the baseline IOP and an IOP at or below a target that was predefined according to the grade of visual field loss. The researchers found no statistically significant difference in the percentage reduction in IOP after the first and second SLT (two-tailed paired t-test). Primary SLT was successful in 55% of eyes, with a mean duration of success of 6.9 (3.4) months. Repeated SLT was successful in 66% of eyes, with a mean duration of success of 13.1 (11.2) months [70]. 

Garg et al. compared IOP values in 115 eyes of 90 patients from the LIGHT study before and 2 months after primary and repeated SLT. The baseline IOP before the initial SLT was significantly higher than IOP before the repeated SLT (mean difference of 3.4 mmHg, 95% CI, 2.6–4.3 mmHg, *p* < 0.001). Although absolute IOP reduction at 2 months was greater after initial SLT compared with repeated SLT (mean difference of 1.0 mmHg, 95% CI, 0.2–1.8 mmHg, *p* = 0.02), the adjusted absolute IOP reduction at 2 months was greater after the repeated SLT (adjusted mean difference 1.1 mmHg, 95% CI, 1.7–0.5 mmHg, *p* = 0.001). This may show the added effect of repeated SLT [31].

Due to different designs, patient characteristics, sample sizes and follow-up periods, the studies evaluating SLT repeatability are not easily comparable. Table 2 summarizes the differences. 

## 9. Complications

SLT is a safe procedure with a very low complication rate most of which are transient and self-limiting. The most frequent side effects mentioned in the systemic review and meta-analysis by Wong et al. were IOP spikes, anterior chamber inflammation, eye pain or discomfort, and peripheral anterior synechiae [28,71].

### 9.1. IOP Spikes

In Wong’s analysis, the frequency of IOP spikes varied from 0 to 28%. In the LIGHT trial, in which 776 SLTs were performed, only 6 IOP spikes were observed, of which only 1 patient required additional treatment. In their case series, Harasymowicz et al. described four cases of IOP elevation up to 46 mmHg in patients with highly pigmented angles. Three of these patients presented features of pigment dispersion syndrome. Two patients had undergone ALT prior to SLT, and one patient had a history of ocular trauma. All patients required medical therapy, and three patients underwent trabeculectomy [13,14,28,72].

### 9.2. Anterior Chamber Inflammation

This adverse event was evaluated as part of the West Indies Glaucoma Laser Study (WIGLS). In this study, cells and flare in the anterior chamber were observed in 40.3% and 9.7% of eyes, respectively. One patient presented bilateral anterior iritis one day after the procedure. The patient had concealed a previous history of recurrent iritis. In this case, signs of iritis were resolved after one week of topical treatment with 1% prednisolone acetate used four times a day. In all other patients, signs resolved without any treatment, as no anti-inflammatory prophylaxis was used in this study. One case of severe iritis with uveal effusion was reported [73,74].

### 9.3. Eye pain, Discomfort and Redness

Eye pain, discomfort and redness are frequently reported adverse events in many clinical trials. These mild, self-limiting symptoms are mainly related to the SLT technique [28,71].

### 9.4. Peripheral Anterior Synechiae

Wong et al. reported PAS in 0-2.86% in 12 studies. Baser et al. reported two cases of PAS after repeated SLT. In both cases, PAS did not progress and IOP remained stable [75].

### 9.5. Corneal Adverse Events

We found cases of corneal edema with corneal endotheliitis, corneal haze with stromal edema and herpetic keratitis. Bettis et al. reported a case series of five XFG patients with an IOP spike, two of whom developed corneal decompensation requiring keratoplasty. Several studies have shown minor transient endothelial changes after SLT, such as a reduction in endothelial cell count and central corneal thickness or an increase in dark spots in specular microscopy [76,77,78,79,80,81].

### 9.6. Other

Other reported rare complications of SLT are hypopyon with possible herpes simplex reactivation, hyphema, cystoid macular edema in patients with diabetes, and post-ocular trauma. One case reported foveal burns resulting from the use of capsulotomy mode during SLT [74,75,76,77,78,82,83,84,85,86].

## 10. Summary

Treating glaucoma is a long-term challenge, with the probability that most patients with newly diagnosed glaucoma will be treated for the duration of their lifetime. With a range of alternatives, such as medication, laser, surgery and minimally invasive glaucoma surgery, clinicians should strive for effectiveness in terms of patient adherence; quality of life; and balanced, cost-effective therapy. Reducing the burden of medical treatment and its complications is the main objective of this struggle.

SLT has proven to be an effective and safe method for reducing IOP in newly diagnosed OAG and OHT patients. The results of the well-designed and effectively conducted LIGHT study raise the question of whether medical therapy should still be proposed as first-line therapy in OAG and OHT. Presumably, it is better to induce medical therapy only in cases of SLT failure, especially for patients with higher baseline IOPs with higher predicted IOP reductions. With an efficacy similar to the primary procedure, repeated SLT offers the valuable possibility of achieving long-lasting drop-free periods in glaucoma treatment. This might be important not only for the reduction in topical medication complications but also with regard to the frequent lack of compliance as well as problems with the application of eye drops, which is particularly troublesome for elderly glaucoma patients.

In patients whose IOP is already controlled with topical treatment, it is possible to reduce or even discontinue medication, which enhances treatment-related quality of life. Patients with uncontrolled IOP on maximally tolerable medication can achieve the target IOP after SLT. Some reports showed partial efficacy of SLT in advanced OAG, which can be important given the high risk of postoperative complications that accompany incisional glaucoma surgery. All of the advantages of SLT described above are also of particular value to young patients, in whom SLT has been shown to be effective and who, at the time of diagnosis, face the prospect of life-long therapy and can benefit the most from delaying the introduction of other treatment modalities.

As an important limitation of the most cited studies (with exception of the LIGHT study), one should mention concentration on the IOP values as the outcome measures, whereas in real-life decision making it is the progression observed in visual field or imaging that implies possible treatment escalation. In this field further well-designed randomized trials are required [87].

## Figures and Tables

**Figure 1 jcm-10-03307-f001:**
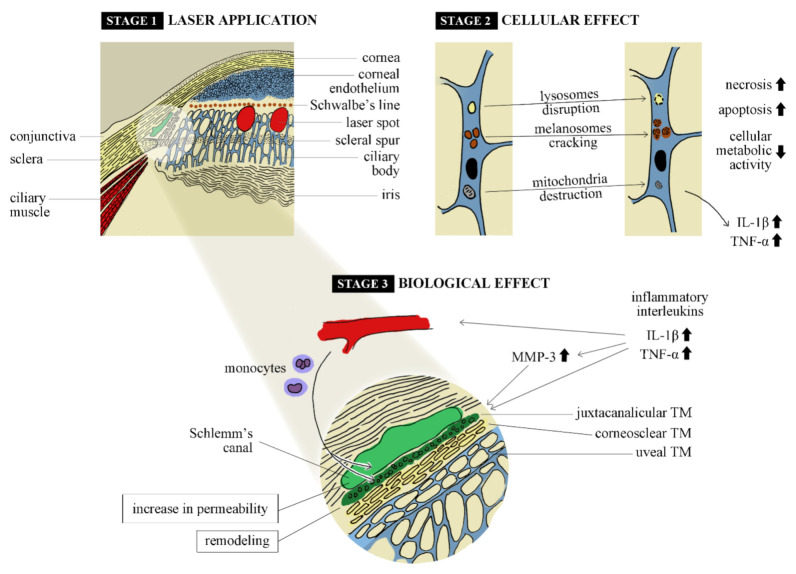
Summary of the trabecular meshwork tissue response induced by selective laser trabeculoplasty.

**Table 1 jcm-10-03307-t001:** Summary of the effectiveness of SLT in relation to the scope of the procedure performed (the number of quadrants of the circuit covered by laser therapy).

Research	Type of Study	Number of Subjects	Diagnosis	Indications	IOP Reduction	Differences in Procedure	Observation Time
Chen et al. 2004 [16]	Prospective, single-masked randomized clinical trial	N_90_ = 32	OAG, OHT	Adjunctive	5.1 mmHg (1 mo)	90 degrees, some patients received ALT	Up to 7 months
6.3 mmHg (4 mo)
6.2 mmHg (7 mo)
	N_180_ = 32	OAG, OHT	Adjunctive	4.2 mmHg (1 mo)	180 degrees, some patients received ALT	Up to 7 months
5.6 mmHg (4 mo)
7.1 mmHg (7 mo)
Goyal et al. 2010 [17]	Prospective, single-masked randomized clinical trial	N_180_ = 18	OAG = 12,	Primary	6.1 mmHg (1 mo)	180 degrees	1 month
OHT = 6
	N_360_ = 19	OAG = 15,	Primary	9.0 mmHg (1 mo)	360 degrees	1 month
OHT = 4

Shibata et al. 2012 [18]	Retrospective cohort study	N_180_ = 35	OAG	Additive	2.6 mmHg (6 mo)	180 degrees	Up to 36 months
	N_360_ = 34	OAG	Additive	5.6 mmHg (6 mo)	360 degrees	Up to 36 months
Francis et al. 2016 [19]	Open-label prospective study	N_360_ = 137	POAG	Additive	4.1 mmHg (6–12 mo)	360 degrees	Up to 15 months
3.5 mmHg (12–15 mo)
Tawfique et al. 2019 [20]	Prospective, single-masked randomized clinical trial	N_90_ = 32	OAG, OHT, XFG	Primary or Additive	nd (only survival analysis was conducted	90 degrees	Up to 24 months
	N_180_ = 35	OAG, OHT, XFG	Primary or Additive	nd (only survival analysis was conducted)	360 degrees	Up to 24 months
Özen et al. 2020 [21]	Open-label prospective study	N_180_ = 26	POAG	Additive	7.8 mmHg (1 mo)	180 degrees	Up to 6 months
9.0 mmHg (3 mo)
9.1 mmHg (6 mo)
	N_360_ = 26	POAG	Additive	8.5 mmHg (1 mo)	360 degrees	Up to 6 months
10.1 mmHg (3 mo)
10.3 mmHg (6 mo)
Nirappel et al. 2021 [22]	Retrospective cohort study	N_180_ = 196	OAG, OHT	Additive	2.9 mmHg (6 wk)	180 degrees	Up to 24 months
3.0 mmHg (12 mo)
2.2 mmHg (24 mo)
	N_360_ = 258	OAG, OHT	Additive	3.2 mmHg (6 wk)	360 degrees	Up to 24 months
3.4 mmHg (12 mo)
2.4 mmHg (24 mo)

mo—months; wk—weeks; OAG—open-angle glaucoma; OHT—ocular hypertension; POAG—primary open-angle glaucoma; nd—no data.

**Table 2 jcm-10-03307-t002:** Summary of studies evaluating SLT repeatability.

Research	Hong et al. in 2009 [68]	Avery et al. in 2013 [70]	Francis et al. in 2016 [19]	Polat et al. 2016 [69]	Garg et al. 2020 [31]
Patient characteristics	POAG, XFG, and PG uncontrolled on medication	Treatment-naïve POAG	Primary and secondary glaucoma (except uveitic) on medication	POAG, XFG and PG uncontrolled on medication	Treatment-naïve OHT and OAG that required repeated SLT
Interval between 1st and 2nd SLT (time in which 1st SLT remained successful)	Not specified	Not specified	At least 6 months	At least 6 months	Maximum 18 months
Sample size	44 eyes of 35 patients	42 eyes of 42 patients	137 eyes of 137 patients	38 eyes of 38 patients	115 eyes of 90 patients
Range of SLT	360°	360°(40–50 spots)	360° (80–132)	360° (100 spots)	360° (100 spots)
Definition of success	≥20% reduction from baseline	IOP reduction of ≥20% of baseline IOP and IOP at or below predefined target	1st definition:	“Real-world” definition: IOP control without additional IOP-lowering medications, glaucoma laser procedures or incisional glaucoma surgery;	IOP at or below target IOP without additional IOP-lowering medications, further laser procedures or incisional glaucoma surgery
			IOP 5–21 mmHg, IOP reduction >20%, no addition of medication or procedure;	“formal” definition IOP reduction ≥20%	
			2nd definition:		
			IOP 5–21 mmHg, no additional glaucoma procedure, either IOP reduction of >20% or reduction in medication		
Success rate of first SLT	50%	55%	1st definition:	Not specified; “real-world” definition:	Not specified;
	(*p* = 0.52)		55% at 6 months and 34% at 12 months;	Kaplan–Meier survival analysis showed a median survival time of 570 days;	
			2nd definition:	“Formal” definition:	Kaplan–Meier survival analysis
			65% at 6 months and 44% at 12 months	Kaplan–Meier survival analysis showed a median survival time of 270 days	showed a median duration of effect of 189 days
Success rate of second SLT	43.2% (*p* = 0.52)	66%	1st definition: 37% at 6 months and 19% at 12 months;	Not specified;	Not specified;
			2nd definition: 48% at 6 months and 27% at 12 months	“real-world” definition:	
				Kaplan–Meier survival analysis showed a median survival time of 1054 days;	Kaplan–Meier survival analysis could not be conducted, as 50% of eyes had not reached the endpoint by the end of follow-up
				“Formal” definition:	
				Kaplan–Meier survival analysis showed a median survival time of 360 days	
Time of evaluation or follow-up period	5–8 months follow-up	Mean duration of follow-up:	12–15 months’ follow-up	up to 24 months	18 months’ follow-up
		1st SLT: mean duration of follow-up was 10.5 months;			
		2nd SLT: 15.1 months;			
		3rd SLT: 9.0 months			
Mean IOP reduction after 1st SLT [mmHg]	4.0 (5.3,2.7) at 5–8 months	3.6 (4.8) at second visit (4–5 months)	4.1 (SD 4.8) at 6–12 months (*p* < 0.001)	2.9–5.7 at different points in 24-month follow-up	5.3 (4.5–6.0) [95% CI] at 2 months
Mean IOP reduction after 2nd SLT	2.9 (4.2,1.5) mmHg at 5–8 months (*p* = 0.16)	4.5 (4.5) at second visit (4–5 months)	2.9 (SD 4.7) at 6–12 months (*p* < 0.001)	2.3–4.4 at different points in 24-month follow-up	4.6 (4.0–5.2) [95% CI] at 2 months

## Data Availability

Not applicable.

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
