# Peer review of "Selective Laser Trabeculoplasty in the Treatment of Ocular Hypertension and Open-Angle Glaucoma: Clinical Review"

_jcm, 2021, doi:10.3390/jcm10153307_

Round 1

Reviewer 1 Report

All my concerns have been properly resolved.

Author Response

Thank you for your kind words.

Reviewer 2 Report

A very comprehensive and elaborate review regarding SLT for OHT and OAG.

I'd only wish that the authors perform a similar review regarding the possible use of SLT also for closed angle glaucoma.

Author Response

We have written a new small section covering possible use of SLT in PAC/PACG.

Reviewer 3 Report

Congratulations

Overall an excellent review of the subject well organised and presented.

These are my comments which can be easily addressed by the authors:

  1. Lines 208-225: An important unique feature of the Light Trial is that treatment protocol allowed de-escalation of treatment in cases where IOP was above target but the patient manifested disease stability.
  2. Lines 208-225: In the Light trial the SLT arm demonstrated as the authors mentioned a smaller incidence of cataract and glaucoma incisional surgeries. Whereas the difference in cataract surgeries between treatment arms is modest the difference in trabeculectomies is vast (0 vs. 11 cases). I believe that this differentiation needs to be specifically addressed in this review.
  3. Line 273: The first report of prostaglandin associated periorbitopathy in the ophthalmic literature is by Filippopoulos et al. 
  4. Lines 614-641: One limitation of the literature on SLT that may be included in this review is the fact that almost all studies (due to insufficient follow-up)  do not report on the effect of SLT on VF stability with the exception of maybe the Light Trial (Wright DM et al. Ophthalmology 2020;127:1313-21​)

Author Response

Lines 208-225: An important unique feature of the Light Trial is that treatment protocol allowed de-escalation of treatment in cases where IOP was above target but the patient manifested disease stability. - Information added in lines 198-201.

We added in the text the information that you indicated:

Target intraocular pressure was reduced by 20% if deterioration was identified despite the measured intraocular pressure being at or below target. If the intraocular pressure was above target by less than 4 mm Hg, but with no evidence of deterioration, then the target intraocular pressure was revised to the mean of the previous three visits over which deterioration had not occurred.

Lines 208-225: In the Light trial the SLT arm demonstrated as the authors mentioned a smaller incidence of cataract and glaucoma incisional surgeries. Whereas the difference in cataract surgeries between treatment arms is modest the difference in trabeculectomies is vast (0 vs. 11 cases). I believe that this differentiation needs to be specifically addressed in this review. - Information added.

Line 273: The first report of prostaglandin associated periorbitopathy in the ophthalmic literature is by Filippopoulos et al.  - link for bibliography added: Filippopoulos T, Paula JS, Torun N, Hatton MP, Pasquale LR, Grosskreutz CL. Periorbital changes associated with topical bimatoprost. Ophthal Plast Reconstr Surg 2008; 24: 302–307.

Lines 614-641: One limitation of the literature on SLT that may be included in this review is the fact that almost all studies (due to insufficient follow-up) do not report on the effect of SLT on VF stability with the exception of maybe the Light Trial (Wright DM et al. Ophthalmology 2020;127:1313-21) - Link for bibliography added: https://pubmed.ncbi.nlm.nih.gov/32402553/ Comment added in summary. 

Reviewer 4 Report

Τhe present paper is  a review of the role Selective Laser Trabeculoplasty in the Treatment of Ocular  Hypertension and Open-Angle Glaucoma. Although there a lot of similar reviews in the literature the addition of some relative novel studies makes this paper worth publishing Some minor comments

  • In this type of paper materials and methods heading does not apply
  • Line 193: in the Laser-first group and 362 patients assigned to the Medicine-first group. Capital letters are not indicated.

Author Response

Line 193: in the Laser-first group and 362 patients assigned to the Medicine-first group. Capital letters are not indicated - corrected in the whole section

In case of Methods section we followed the Preferred Reporting Items for Systematic review and Meta-Analysis Protocols (PRISMA protocol, http://www.prisma-statement.org/) as closely as we could in order to give the readers as much objective and synthetic analysis as we could. In order to fulfil this requirement we described the modes and methods how we search for knowledge. Due to this same PRISMA protocol we prepared full search strategy report in supplementary data to be as transparent as we could and for our effort to be easily verified and replicated if the novel revision will be necessary.

This manuscript is a resubmission of an earlier submission. The following is a list of the peer review reports and author responses from that submission.

Round 1

Reviewer 1 Report

The authors examined the literature published so far and summarize the effects of Selective Laser Trabeculoplasty (SLT) on glaucoma patients in an easy-to-understand manner. They started with mechanisms of action of SLT and analyzed the difference in intraocular pressure due to laser irradiation range and number of postoperative eye drops. They also compared the therapeutic effects of untreated glaucoma and those treated with eye drops and concluded that good intraocular pressure control was obtained in both cases. They also summarized differences by disease type and age. I don't see any points that should be greatly modified in this review.

Minor points:

In Figure 1 (P4),

Ciliary body band should be ciliary band.

Ciliary body muscle should be ciliary muscle.

Author Response

Rewiever 1: Comments and Suggestions for Authors

The authors examined the literature published so far and summarize the effects of Selective Laser Trabeculoplasty (SLT) on glaucoma patients in an easy-to-understand manner. They started with mechanisms of action of SLT and analyzed the difference in intraocular pressure due to laser irradiation range and number of postoperative eye drops. They also compared the therapeutic effects of untreated glaucoma and those treated with eye drops and concluded that good intraocular pressure control was obtained in both cases. They also summarized differences by disease type and age. I don't see any points that should be greatly modified in this review.

Answer: We want to thank the reviewer for great effort in correcting and reading of our manuscript.

Minor points:

In Figure 1 (P4),

Ciliary body band should be ciliary band

Answer: We modified the figure. The description was corrected.

Ciliary body muscle should be ciliary muscle

Answer: We modified the figure. The description was corrected.  

Reviewer 2 Report

 A review on SLT is undobtely interesting but   there are some pitfall in this text:

1- some part are not clear and easy to understand (e.g. lines 59-63 and lines 98-102)

2- legend of fig .1 is not clarifyng

3- some spelling error (e.g. line 233 preformed instead of performed.

4- line 349 consider if adiunctive therapy is better than adjacent therapy

5- why not something more on pigmentary glaucoma?

Finally the text has to be revised in order to be more fluent

Author Response

Reviewer 2: Comments and Suggestions for Authors

 A review on SLT is undobtely interesting but   there are some pitfall in this text:

  • some part are not clear and easy to understand (e.g. lines 59-63 and lines 98-102)

Answer: Whole section “Mechanisms of action” was re-written.        

2- legend of fig .1 is not clarifying –

Answer: Figure 1. And its descriptions were corrected.

3- some spelling error (e.g. line 233 preformed instead of performed.

Answer: This error was corrected.

4- line 349 consider if adjunctive therapy is better than adjacent therapy

Answer: We have corrected this description in the whole text.

5- why not something more on pigmentary glaucoma

Answer: We have written the whole new subsection about pigmentary glaucoma.

Finally the text has to be revised in order to be more fluent

Answer: The text was revised and corrected by academic, professional proof-reader Richard Peters.

We want to thank the reviewer for pointing out important deficits that our previous version of manuscript bare. We hope that the revised version would be worth of being published in such important journal.

Reviewer 3 Report

1) Stick to primary open-angle glaucoma or open-angle glaucoma, and hyphenate as appropriate (and follow hyphenation rules through the manuscript - consistency matters).

2) The abstract is using semi-colons, when it should use commas. ", and it helps..."

3) with "a" specific protocol?

4) Unlike in previously performed argon laser trabeculoplasty (ALT), thermal damage to adjacent nonpigmented structures is avoided with SLT - please have a native English speaker review for flow.

5) A randomized clinical trial is the highest level of evidence, so that sentence at the end of the abstract sounds awkward. Also, "should do X" sounds too authoritarian. "We suggest X" sounds more like a review article...

6) Calling SLT a basic treatment is awkward - reword. 

7) Attached file? I would just say it's in the Supplementary Materials.

8) -and safety => extra hyphen where it doesn't belong

9) the-therapeutic => extra hyphen where it doesn't belong

10) First sentence of MOA section is awkwardly written => again, native English speaker should re-read entire article to avoid awkwardly written sentences.

11) I would just call it Schlem's canal, not just canal. I wonder if a picture describing the 7 and 14 micron descriptions would help.

12) WHat's the mechanical effect? How did the Kramer study exclude it? This needs a better explanation.

13) The Bradley study paragraph starts mentioning molecules that I don't think the average glaucoma specialist is well versed in. If you're going to mention them, explain what they are how they tie to the paragraphs above. 

14) The whole MOA section needs to be re-written. It's wordy and the paragraphs and ideas don't flow into each other. Instead of organizing by study, maybe organize by ideas. If you keep the study organization, maybe a figure to show a timeline of when each part of the MOA was discovered?

15) I like the colors in figure 1, BUT maybe have steps (1), (2), (3)... the arrows are confusing. WHat happens first, second? WHat are these molecules and why do they matter?

16) Technique: Some of us start at an energy of 0.3 or 0.4 until bubbles are seen. I actually like to see bubbles 50% of the time, so I disagree with "minimal" bubble formation. What does Dr. Latina describe? What have studies described? I think the energy levels are open for debate. There's quite a bit of debate how many degrees to treat. The article you cite in the table by Nirappel et al explains this. Some people even treat 90 degrees. In the Table, I wouldn't say "Impacts on," just say number of degrees...

Please re-read the entire manuscript with the help of a native English-speaker. Once the organization has been improved and the syntax/flow has been dramatically improved, I can read again.

Author Response

Reviewer 3: Comments and Suggestions for Authors

  • Stick to primary open-angle glaucoma or open-angle glaucoma, and hyphenate as appropriate (and follow hyphenation rules through the manuscript - consistency matters).

Answer: The name was corrected in the whole text.

2) The abstract is using semi-colons, when it should use commas. ", and it helps..." –

Answer: Abstract was corrected by professional proof-reader.

3) with "a" specific protocol?

Answer: Articles were corrected by professional proof-reader.

4) Unlike in previously performed argon laser trabeculoplasty (ALT), thermal damage to adjacent nonpigmented structures is avoided with SLT - please have a native English speaker review for flow.

Answer: Sentence was corrected by professional proof-reader.

5) A randomized clinical trial is the highest level of evidence, so that sentence at the end of the abstract sounds awkward. Also, "should do X" sounds too authoritarian. "We suggest X" sounds more like a review article... –

Answer: We corrected the nouns.

6) Calling SLT a basic treatment is awkward - reword.

Answer: Sentence was corrected by professional proof-reader. “Basic” was changed to “essential”.

7) Attached file? I would just say it's in the Supplementary Materials.

Answer: Name was corrected adequately to suggestion.

8) -and safety => extra hyphen where it doesn't belong

Answer: Hyphens were corrected by professional proof-reader.

9) the-therapeutic => extra hyphen where it doesn't belong

Answer: Hyphens were corrected by professional proof-reader.

10) First sentence of MOA section is awkwardly written => again, native English speaker should re-read entire article to avoid awkwardly written sentences.

Answer: Whole section was re-written. Entire article was revised and corrected by professional, academic proof-reader Richard Peters.

11) I would just call it Schlem's canal, not just canal. I wonder if a picture describing the 7 and 14 micron descriptions would help. –

Answer: This part was excluded from the new “Mechanisms of action” section.

12) What’s the mechanical effect? How did the Kramer study exclude it? This needs a better explanation.

Answer: We have clarified mechanical effect and why it is not a case in SLT in new “Mechanisms of action” section.

13) The Bradley study paragraph starts mentioning molecules that I don't think the average glaucoma specialist is well versed in. If you're going to mention them, explain what they are how they tie to the paragraphs above.

Answer: We have clarified this in the new “Mechanisms of action” section.  

14) The whole MOA section needs to be re-written. It's wordy and the paragraphs and ideas don't flow into each other. Instead of organizing by study, maybe organize by ideas. If you keep the study organization, maybe a figure to show a timeline of when each part of the MOA was discovered? –

Answer: Whole section was re-written and organized by ideas.

15) I like the colours in figure 1, BUT maybe have steps (1), (2), (3)... the arrows are confusing. WHat happens first, second? What are these molecules and why do they matter?

Answer We have rebuilt Figure 1., divided to stages and added new descriptions. Role of molecules is detailed in main text.

16) Technique: Some of us start at an energy of 0.3 or 0.4 until bubbles are seen. I actually like to see bubbles 50% of the time, so I disagree with "minimal" bubble formation. What does Dr. Latina describe? What have studies described? I think the energy levels are open for debate. There's quite a bit of debate how many degrees to treat. The article you cite in the table by Nirappel et al explains this. Some people even treat 90 degrees. In the Table, I wouldn't say "Impacts on," just say number of degrees – corrected

Answer: We have changed the energies according to the studies that we analysed. We commented on bubble formation, according to suggestion. Comparison of range of TM treated is shown in Table 1. “Impacts on” was deleted from the table.

Please re-read the entire manuscript with the help of a native English-speaker. Once the organization has been improved and the syntax/flow has been dramatically improved, I can read again. –

Answer: Entire article was revised and corrected by professional, academic proof-reader Richard Peters.

Round 2

Reviewer 2 Report

carefully review english language: some examples

p.3 assesed;  p.7 linaprive (Linaprive); p.8 Except (or Besides?); p.8 and anxiety and depression ( , anxiety and depression);p 14 apart from in (?); p 15 -16 repeat SLT

Author Response

Number of changes improving english grammar and syntax has been made.    Line 192: "Additionally," added  Line 194:  "In this study's in vitro experimentation human monocytes..." Line 202: enter at the end of the section  Line 214: "assessed" instead of assesed"  Line 261: "exfoliative glaucoma" instead of "Exfoliative Glaucoma"  Line 332: "Besides reductions...  Line 335: "reducing IOP to target values in patients..."  Line 432: "as well as" instead of "and"  Line 511: "treatment" erased  Line 512: "may" erased  Line 514: "an option"  Line 523: "personally" erased  Line 580: "t" erased  Line 648: "The XFG and POAG groups consisted of 114 and 142 patients"  Line 685: "repeated SLT"  Line 695: space erased  Line 722: "maximally-tolerable"  Line 787-788 "in cases with" erased  Line 808, 818, 820, 821, 825, 826, 827, 841, 842, 843, 862, 864, 866: "repeated SLT" 
Line 864: space erased  Table 2.2. some spaces between numbers and "%" erased 

Reviewer 3 Report

I had to stop reading again after page 5 - I think this is important content, and I want them to get this published, BUT Mr. Peters or whoever they chose to help them with English and organization has to spend more time with the authors making sure basic essay structure is followed. As it is, it's TOO HARD to read. If JCM allows it, a Word version can be sent to me next time, and I can make all of my comments and corrections directly on the manuscript with track changes. I want to be able to comment on the ideas presented, not so much on the writing, but I need to comment on the writing, it's easier to do so on a Word document.   1) Abstract: "... when used as primary treatment,..." This sentence isn't clear.  2) Abstract: "...that may delay"  3) Why not abbreviate to SLT at the beginning and then use SLT throughout abstract? Same thing for IOP? It may make it easier to read. 4) Abstract: "...only when laser procedures have failed..." I suggest Mr. Peters re-reads the abstract. It's improved, but it's still not perfectly written. 5) Introduction: "For this reason,..." Have Mr. Peters revise commas, semi-colons, colons throughout 6) "a specific protocol" - missing an "a," but this sounds awkward. 7) "most types of glaucoma", right? Isn't POAG the most common type? 8) The paragraph that starts "Taking into account..." is wordy and awkwardly written. 9) The sentence "The aim of the paper" is too wordy. 10) "Its exact mechanisms have​..." Again, Mr. Peters needs to carefully read for grammar mistakes. 11) Several studies have shown possible... instead of "several studies conducted have proved" 12) Comma after 1995 13) Applied "to" the TM cells 14) As "a" chromophore 15) The paragraph is improved, but it still reads in a stilted fashion/manner. Re-word "Due to exposure times..." 16) Can we just use TM as an abbreviation for "trabecular meshwork"? It may help make some parts more readable. 17) What are "ablation craters"? Again, let's make sure each sentence is teaching something meaningful to the readers 18) The sentence "Further study..." sounds stilted as well. "compare" A to Z - it's too long. 19) with "the" work of Kramer et al. 20) I love the explanation about champagne bubbles, but how does it fit into the rest of the paragraph? 21)  Why are we describing the mechanical theory of ALT? Isn't the review article about SLT? If you insist on including this to contrast it to SLT, transition sentences are needed in between paragraphs.  22) Saying that no mechanical effects whatsoever occur with SLT seems farfetched. Maybe the sub-headings for Mechanism of Action can be "Mechanical Effect Theory" and "Biological Effect Theory"  23) Transition into the Biological Changes section better.  24) "...the broadest..." - sentence is a bit long. 25) What are "anterior segment explants"? Cadaver eyes? Do you mean aqueous humor? 26) Now we're talking about ALT with Bradley et al? If the review article is focused on SLT, focus on SLT. Only bring ALT into the discussion if it's to contrast against SLT in a meaningful way. 27) If you're going to indicate years for studies, be consistent and do it for all like "Lee et al in 2016" 28) For the studies in the biological effect section, it seems as if a sentence or two was taken from each study and placed into the paragraph. Sentences need to flow, and each paragraph needs to carry a message/tell a story that connects and transitions well into the messages/stories other paragraphs carry/tell. 29) I would avoid discussing prostaglandin analogues... The writing is confusing enough as is. 30) I don't understand the message of the paragraph that starts with Goyal et al in 2010. If it makes sense, maybe start with the older studies, and then see how the newer studies build on the knowledge from the earlier studies. The section on MOA is disorganized - see point #28. 31) "to develop this treatment alternative"? What does that mean? Also, based on your MOA section, what's missing in the research into the MOA. You make it sound as if we don't know anything, but there's a lot we know. Maybe a paragraph explaining what we don't know and how knowing it would help clinically would be helpful.  32) Figure 1 is still confusing. At first, the circle seemed to be the actual laser application, and it took me a minute to realize it wasn't. If you're going to label it "laser application," then show the laser application. For me cellular and biological are essentially equivalent... the arrow goes from Stage 1 to Stage 3... The colors are beautiful, but more thought must be given to the organization of this figure and what you're going to show at each stage... I would also make sure that each stage is explained in the text above. Figure 1 can serve as a true summary of the MOA section. Maybe that can help focus/tighten the section AND Figure 1 at the same time. 33) Is "champagne bubble" the most scientific term for what we see. I'm OK with you using it, but then stick to it throughout the text. I would say that we should use the term that is most used in the literature.  34) "until" not till.... also "some percent of the spots" sounds awkward. Again, Mr. Peters needs to re-read this paragraph and edit for flow. 35) What about the studies that show 90-degrees of treatment is enough? 36) I would probably organize the table by date of study. Start with oldest and finish with newest. 37) single-masked (hyphen needed!) 38) "Non" in the table - what? Are OAG and POAG the same in the table? 39) "some patients received ALT" in lieu of "in some patient ALT"...  40) I routinely do not give any postoperative steroids/NSAIDs to my SLT patients... Maybe Table 1 can include postoperative regimens when available?  41) Personally administer personally?   I have to stop correcting.

Author Response

We corrected all comments indicated by the reviewer according to the list (changes indicated by italics)

Comments and Suggestions for Authors

I had to stop reading again after page 5 - I think this is important content, and I want them to get this published, BUT Mr. Peters or whoever they chose to help them with English and organization has to spend more time with the authors making sure basic essay structure is followed. As it is, it's TOO HARD to read. If JCM allows it, a Word version can be sent to me next time, and I can make all of my comments and corrections directly on the manuscript with track changes. I want to be able to comment on the ideas presented, not so much on the writing, but I need to comment on the writing, it's easier to do so on a Word document.

  • Abstract: "... when used as primary treatment,..." This sentence isn't clear.

Sentence is divided to shorter parts.

  • Abstract: "...that may delay"

This part is corrected.

  • Why not abbreviate to SLT at the beginning and then use SLT throughout abstract? Same thing for IOP? It may make it easier to read.

We thought, we cannot use abbreviations in abstract. All abbreviations are corrected.

  • Abstract: "...only when laser procedures have failed..." I suggest Mr. Peters re-reads the abstract. It's improved, but it's still not perfectly written.

This part is corrected.

  • Introduction: "For this reason,..." Have Mr. Peters revise commas, semi-colons, colons throughout.

Commas are corrected.

  • "a specific protocol" - missing an "a," but this sounds awkward.

We erased this part.

  • "most types of glaucoma", right? Isn't POAG the most common type?

This part is corrected.

  • The paragraph that starts "Taking into account..." is wordy and awkwardly written.

This part was re-written.

  • The sentence "The aim of the paper" is too wordy.

The sentence was shortened.

  • "Its exact mechanisms have​..." Again, Mr. Peters needs to carefully read for grammar mistakes.

This mistake is corrected.

  • Several studies have shown possible... instead of "several studies conducted have proved"

This part is corrected.

  • Comma after 1995 – corrected
  • Applied "to" the TM cells – corrected

  • As "a" chromophore - corrected

  • The paragraph is improved, but it still reads in a stilted fashion/manner. Re-word "Due to exposure times..." –

Whole MOA section was re-written.

  • Can we just use TM as an abbreviation for "trabecular meshwork"?

Of course, Abbreviations are corrected.

  • What are "ablation craters"? Again, let's make sure each sentence is teaching something meaningful to the readers

Now it is clarified in the text.

  • The sentence "Further study..." sounds stilted as well. "compare" A to Z - it's too long.

Sentence is re-written.

  • with "the" work of Kramer et al. – corrected

  • I love the explanation about champagne bubbles, but how does it fit into the rest of the paragraph?

This part is erased.

  • Why are we describing the mechanical theory of ALT? Isn't the review article about SLT? If you insist on including this to contrast it to SLT, transition sentences are needed in between paragraphs.

This part is erased.

  • Saying that no mechanical effects whatsoever occur with SLT seems farfetched. Maybe the sub-headings for Mechanism of Action can be "Mechanical Effect Theory" and "Biological Effect Theory"

We changed this information. Whole MOA section is re-written.

  • Transition into the Biological Changes section better.

This is corrected in the new version of MOA section.

  • "...the broadest..." - sentence is a bit long.

Sentence is re-written.

  • What are "anterior segment explants"? Cadaver eyes? Do you mean aqueous humor?

Now it is better clarified in the text.

  • Now we're talking about ALT with Bradley et al? If the review article is focused on SLT, focus on SLT. Only bring ALT into the discussion if it's to contrast against SLT in a meaningful way.

This is also possible mechanism of SLT. We clarified it in the text.

  • If you're going to indicate years for studies, be consistent and do it for all like "Lee et al in 2016"

We erased years from the text.

  • For the studies in the biological effect section, it seems as if a sentence or two was taken from each study and placed into the paragraph. Sentences need to flow, and each paragraph needs to carry a message/tell a story that connects and transitions well into the messages/stories other paragraphs carry/tell.

Whole MOA section is re-written.

  • I would avoid discussing prostaglandin analogues... The writing is confusing enough as is.

We erased this part.

  • I don't understand the message of the paragraph that starts with Goyal et al in 2010. If it makes sense, maybe start with the older studies, and then see how the newer studies build on the knowledge from the earlier studies. The section on MOA is disorganized - see point #28.

We erased this part

  • "to develop this treatment alternative"? What does that mean? Also, based on your MOA section, what's missing in the research into the MOA. You make it sound as if we don't know anything, but there's a lot we know. Maybe a paragraph explaining what we don't know and how knowing it would help clinically would be helpful.

We erased this sentence as it was confusing.

  • Figure 1 is still confusing. At first, the circle seemed to be the actual laser application, and it took me a minute to realize it wasn't. If you're going to label it "laser application," then show the laser application. For me cellular and biological are essentially equivalent... the arrow goes from Stage 1 to Stage 3... The colors are beautiful, but more thought must be given to the organization of this figure and what you're going to show at each stage... I would also make sure that each stage is explained in the text above. Figure 1 can serve as a true summary of the MOA section. Maybe that can help focus/tighten the section AND Figure 1 at the same time.

We changed Figure 1 to show that the circle in STAGE 3 is magnification of Schlemm’s canal and surrounding structures that are visible in STAGE1. Laser spots are marked with red colour in STAGE 1. We changed sizes of STAGE 2 and 3 to show that they are equally important. The text of MOA section was corrected to explain mostly the things shown in Firuge 1.

  • Is "champagne bubble" the most scientific term for what we see. I'm OK with you using it, but then stick to it throughout the text. I would say that we should use the term that is most used in the literature.

We changed this expression to simple “bubble”

  • "until" not till.... also "some percent of the spots" sounds awkward. Again, Mr. Peters needs to re-read this paragraph and edit for flow.

Both things are corrected

  • What about the studies that show 90-degrees of treatment is enough?

We found another study comparing 360 vs. 90 degrees and it was added to the text unfortunately due to lack in the source paper exact post-SLT IOP values (the authors conducted survival analysis and data on raw IOP summary was not available) it partially has been added in the Table 1.

  • I would probably organize the table by date of study. Start with oldest and finish with newest. - corrected

  • single-masked (hyphen needed!) – corrected

  • "Non" in the table - what? Are OAG and POAG the same in the table? –

We corrected the table. The OAG and the POAG are not this same where in the source paper is was possible to distinguish how many of included patients were in each group it was indicated, if it was not possible the information is combined as in the source article.

  • "some patients received ALT" in lieu of "in some patient ALT"...

Corrected

  • I routinely do not give any postoperative steroids/NSAIDs to my SLT patients... Maybe Table 1 can include postoperative regimens when available?

We changed “Postoperative treatment” to show that this treatment is not obvious..

  • Personally administer personally? I have to stop correcting.

This was corrected to “self-administer”

Round 3

Reviewer 3 Report

The writing is still not well organized. A native English speaker who has experience with glaucoma treatments should read it and re-write it.